# Non-Magnetic Circulator Based on a Time-Varying Phase Modulator

Xiangjian Jia and Yanfeng Jiang *

Department of Electrical Engineering, School of IoT Engineering, Institute of Advanced Technology, Jiangnan University, Wuxi 214122, China
* Correspondence: jiangyf@jiangnan.edu.cn

**Abstract:** Non-reciprocal devices are key elements in modern wireless communication systems. The circulator devices can simultaneously save spectrum resources and antennas. A traditional circulator is made of ferrite materials with an external magnetic bias field, and its bulk and incompatibility with CMOS technology can hardly satisfy the miniaturization and integration of modern high-speed communication systems. In recent years, there have been many outstanding achievements in the study of non-magnetic circulators, among which, the method of producing non-reciprocity by temporal modulation is considered the most likely to have a transformative influence on the industry. By varying one of the parameters of the system with time, the time inversion symmetry of the system can be broken so that the non-reciprocal devices can be formed by applying appropriate topological structures without the use of magnetic materials. In the paper, a new concept of a time-varying phase modulator (TVPM) is proposed to achieve a relatively simple method to break the symmetry of time inversion. Two different time-varying phase modulators and buffering units can be integrated to form a gyrator, with which a circulator can be formed. This paper provides a relatively simple design idea and shows the circuit design and implementation method as well as the numerical analysis and simulation results. The simulation results show that the insertion loss of the circulator at the center frequency is −1.7 dB and the isolation is −18 dB. The proposed non-magnetic circulator shows potential applicability in related 5G and pre-6G systems.

**Keywords:** circulator; gyrator; non-reciprocal; non-magnetic; high-speed communication





## 1. Introduction

Reciprocity is an inherent property of a symmetric linear time-invariant system [1]. If one system features a symmetric dielectric tensor or permeability tensor, it implies that this system satisfies Lorentz reciprocity. However, in many modern high-speed wireless communication systems or optical systems, non-reciprocal devices (such as isolators, circulators, and gyrators) play indispensable roles [2–5]. In the full-duplex communication system [6], the circulator is used as a three-port device for communication among the antenna, receiver, and transmitter. It allows the transmitter and the receiver to share the same antenna, and the carrier wave operates at the same frequency band. This strategy can save valuable spectrum resources for the communication system. Traditional circulators are generally composed of ferrite magnetic materials with a permanent magnet used to generate a propagation environment with an external magnetic field [7]. In the circulator with ferrite materials, the electron spin introduces the angular momentum with time inversion asymmetry, resulting in the asymmetric permeability tensor of the system. The asymmetric performance does not satisfy the Lorentz reciprocity theorem and produces a non-reciprocal propagation. However, due to its shortcomings, such as high production cost, bulky size, poor compatibility with the CMOS process, etc., it cannot meet the requirements of modern communication systems [8]. In recent years, many breakthroughs were achieved in the use of non-magnetic nonreciprocal devices. Among them, temporal modulation [9–14] is one of the most promising applicable approaches.

Temporal modulation involves varying one of the parameters of the system with time, which breaks the symmetry of time inversion. The transmission path is changed using a single-pole double-throw switch [4]. The phase of the transmitted signal was changed by using varactors in [11] to change the conductivity of the transmission line, which modulates the phase of the transmitted signal. The frequency as well as the phase of the transmitted signal can be modulated and demodulated by [13] using two mixers on the left and right terminals. All of these methods allow for a change in one of the variables in the structure to break the time-reversal symmetry. However, there is no topology to be concluded with the varied structures. Among the other published methods on time-varying modulations, many similarities can be found. As the N-Path filter [15] shown in Figure 1a, the transmitted carrier waves with different phases can be shifted in propagating and opposite directions based on controlling the switch on/off at different times (Figure 1a). The switching transmission line [16] is the spatiotemporal conductivity modulation, by which the phase reference point can be altered, which can be used to change the carrier wave phase. The transmission line also makes the carrier undergo different phase changes in propagating and opposite directions, resulting in non-reciprocity (Figure 1b). The structure of realizing an optical isolator with a series phase modulator in Reference [17] is similar to the mentioned one above. The sine wave is used as an exciting phase modulator. A delay unit is added to cancel the phase of the transmission wave in a certain direction for realizing one-way transmission (Figure 1c). By summarizing the above three kinds of non-reciprocity circuits, a systematic structure can be extracted to achieve non-reciprocity. As shown in Figure 1d, periodic time-varying modulators are placed on both sides, while buffer stages are placed in the middle stage. The appropriate combination can construct a non-reciprocal circuit. Compared with the traditional circulator composed of ferrite materials, the circulator based on the time-varying modulation method is compatible with CMOS technology. Its size small allows it to be integrated into the integrated circuit (IC).

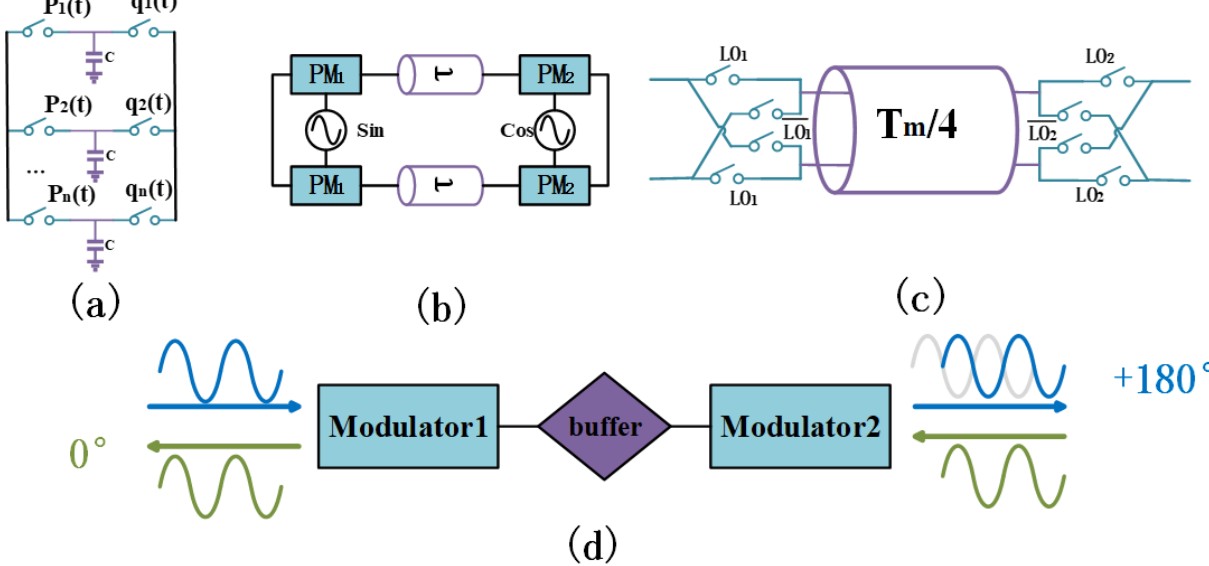

**Figure 1.** Different types of gyrators or isolators and summary topology. (**a**) N-path filter. (**b**) Optical isolator based on tandem modulators. (**c**) Conductivity modulation. (**d**) The general structure of nonreciprocal devices.

Inspired by previous studies, the concept of the time-varying phase modulator (TVPM) is proposed in the paper. The TVPM can be used to modulate the phase of a carrier wave at different times for breaking time-reversal symmetry. A gyrator is composed of two different time-varying phase modulators and appropriate buffering units. The proposed topological structure is used to construct a gyrator. The proposed gyrator and several reciprocal devices [18] can be combined to construct the circulator. Compared with the

gyrator circuit composed of an N-path filter in Reference [15], a larger bandwidth can be achieved with the proposed gyrator. Compared with the gyrator in the method of conductivity modulation in Reference [16], the proposed gyrator in the paper reduces one degree of freedom, which makes the volume smaller and the application involves fewer inductors.

## 2. Gyrator Design

The gyrator is an essential part of the circulator, which is a non-reciprocal two-port device [19]. As shown in Figure 1d, the circulator can make a phase difference of 180° between the propagating and opposite transmission of the signal. In this section, the new concept of TVPM is present, which is used to break the time-reversal symmetry. The demands of the control signal in the composed gyrator circuit are investigated. The two unavoidable losses in the composed gyrator circuit using the TVPM are studied.

### 2.1. Time-Varying Phase Modulator (TVPM)

If a single unit of the phase shifting is placed on a single transmission path, it is a linear time-invariant system. In the system, the transmission of the signal satisfies the reciprocity and the time-reversal symmetry. By connecting two different phase shifters in parallel, when the transmitted carrier wave passes through different phase shifters at different times, the time-reversal symmetry of the phase shifting system can be broken.

The principle of the time-varying phase modulator is shown in Figure 2a. The cylindrical green and purple parts are the units of the actual phase modulator, which can, respectively, change the phase of the transmission wave with $\varphi_A$, $\varphi_B$ and the change of phase varies with time. The signal controlling the rotating device can be used to change the modulation time of the phase modulator. The function of the TVPM acting on the phase of the carrier wave can be written as:

$$\varphi(t) = \left\{ e^{j\varphi_A}, e^{j\varphi_B} \right\} \tag{1}$$

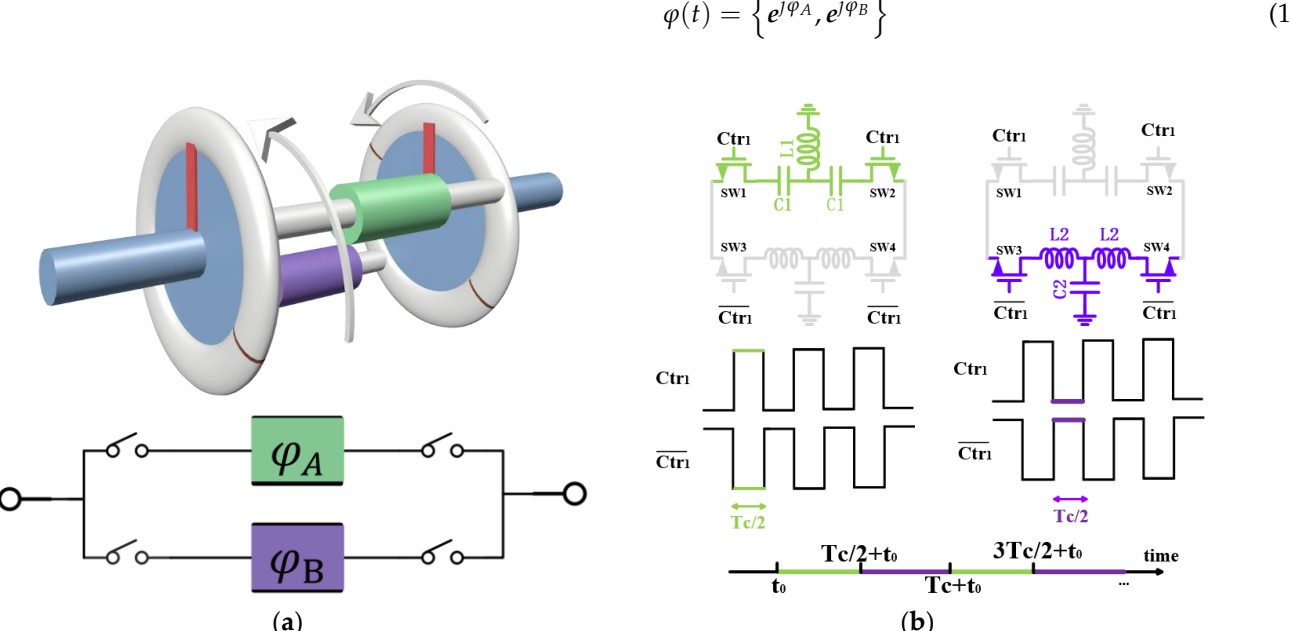

**Figure 2.** Structure of the proposed TVPM. (**a**) The diagram of the proposed TVPM. (**b**) Circuit realization method and key parameters of the proposed TVPM.

The duration of the phase shift of the carrier wave with a single TVPM can be determined by the controlling signal and the magnitude of the phase, which makes the phase modulator more flexible.

Figure 2b shows a circuit realized by a time-varying phase modulator. The phase-shifting unit is composed of switches and lumped elements. The TVPM can be constructed

based on the elements. The circuit is a three-platform T-type 180° high/low pass phase shifter in which C1 = C2 = 3.18 pF, L1 = L2 = 7.96 nH. Notably, it is still a structurally symmetric reciprocity device but breaks the time-reversal symmetry. Using NMOS as the switch, the control signal can be loaded on the gate of NMOS to control the port. During a single period, the control signal has only two states (high and low). When the switch is on at a high level, the signal can be passed. When the switch is off at a low level, the signal is completely reflected.

As shown in Figure 2b, the two control signals are Ctr and $\overline{\text{Ctr}}$, which are inverse to each other. The two signals are loaded on the NMOS gates of the upper and lower arms, respectively. When a signal is transmitted in the first half period (green part), Ctr is at a high level and $\overline{\text{Ctr}}$ is low. So, the switches SW1 and SW2 are off, and SW3 and SW4 are on. The signal can pass through the upper arm of the TVPM. Similarly, in the second half period (purple part), the signal can pass through the lower arm. The upper and lower channels can be opened alternatively at different times. The upper and lower channels can generate 90° and −90° phase shifts for the 1 GHz sine wave, respectively. During one period of the control signal, the upper channel in the first half period is switched on, while the lower channel is off. At this time, the phase change of the carrier wave is determined by the low-pass network. Similarly, the phase change of the carrier wave in the second half cycle is determined by the high-pass network in the lower channel. The phase change of the carrier wave in the first half period is −90°, and the phase change in the second half period is 90°. In addition, the phase-shifting element of the differential construction guarantees a considerable bandwidth for the composed gyrator, which will be described further in the following.

### 2.2. Circuit of Gyrator

The phase-shifting unit can be operated instantaneously without any additional frequency. To realize the function of the gyrator, it is necessary to ensure that the phase change after the output is 180° when the carrier wave enters in different directions. It is assumed that the phase-shifting units of the two TVPM phase-shifting functions $\varphi_1(t)$, $\varphi_2(t)$ are the same, as shown in Figure 3. The phase-shifting of each TVPM can be $\varphi_A$, $\varphi_B$. The difference lies in the phase-shifting of one TVPM being delayed by the other, with the delay time of the buffering unit $\tau$.

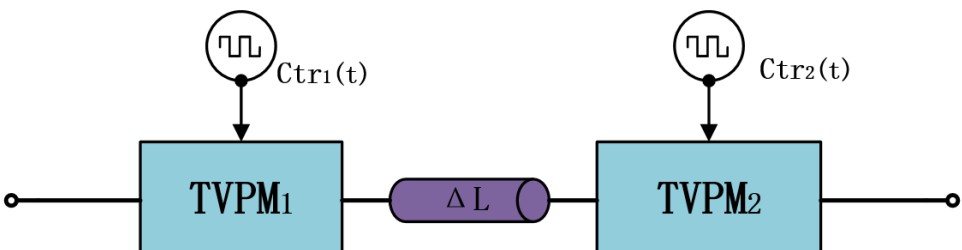

**Figure 3.** Structure of gyrator.

The total phase change of the input signal is affected by the delay transmission line, which acts as the buffering unit. However, this section explores the difference between the phase change of the propagating transmission and the opposite transmission. Therefore, unless otherwise mentioned, the phase change of the signal caused by the intermediate buffering unit is ignored in the analysis of the gyrator in the following. The relationship between the functions of the two TVPMs for phase modulation is expressed by the following:

$$\varphi_1(t) + \varphi_2(t - \tau) = \varphi_2(t) + \varphi_1(t - \tau) \pm \pi \quad \tau = \Delta L / v_g \tag{2}$$

The left side of Equation (2) is the phase change caused by the propagating transmission of the gyrator, while the right side is the phase change caused by the opposite transmission, with a phase difference of 180°.

The nonlinear multivariate equation can be solved by the numerical analysis method. The period Tc controlling of the TVPM signal can be normalized into unit 1. Phase shift $\varphi_A, \varphi_B$ are 0, 1, separately. The periods of the control signals are the same, and both are periodic signals with a duty cycle of 50%. The two TVPMs operating the phase transformation of the carrier wave at different times are:

$$\varphi_1(t) = \varphi_2(t - t_1) = \{0, 1\} \tag{3}$$

$t_1$ is the delay time caused by the buffering unit between the right and left TVPM. It is advisable to set the *PWM* signal function with a duty cycle of 50%, amplitude of 0,1, and initial value of 0 as *PWM* (*t*); the equation after subtraction is as follows:

$$PWM(t) + PWM(t - \tau - t_1) = PWM(t - t_1) + PWM(t - \tau) \pm 1 \tag{4}$$

Four sets of solutions are finally solved, as shown in Table 1:

**Table 1.** Solutions of Equation (4).

| $\tau$ | $t_1$ |
|---|---|
| 0.25 | 0.25 |
| 0.25 | 0.75 |
| 0.75 | 0.25 |
| 0.75 | 0.75 |

When $\tau = t_1 = 0.25$, the delay of the buffering unit and the delay of the right to left TVPM are one-quarter of the controlling signal period *Tc*. When the carrier wave is propagating, as shown in Figure 4a, it is assumed that the carrier wave enters the port in the first half period of the control signal and it enters the right arm TVPM after *Tc*/4. Currently, the change of the carrier phase on the right TVPM is $\varphi_A$ as same as that on the left ones. Therefore, the total phase change of the carrier wave from the port is $2\varphi_A$. When the carrier enters the port in the second half period of the control signal, the carrier output after *Tc*/4 and the total phase change is $2\varphi_B$. When the carrier is transmitted in the opposite way, as shown in Figure 4b, compared with the analysis of forward transmission, the total phase change of the carrier in the output port of the first and second half periods of the control signal is $\varphi_A + \varphi_B$.

Above all, if $\varphi_A - \varphi_B = \pi$, the circuit can achieve the performance of the gyrator. The phase difference between the two carriers transmitted in different directions is $\pi$. The other three sets of solutions are the same as the above analysis, and all can achieve the function of the gyrator. The 180° differential phase shifter mentioned above can be used to construct the TVPM, and the circuit structure of the gyrator can be further obtained.

Figure 5 shows the circuit structure of the gyrator. The TVPM of the left and right arms are differential 180° phase shifters with the same structure. The control signals of the on/off switch are different. The control signal ctr1 is ahead of that of ctr2. The intermediate buffering unit is the delay line, and the delay time is one-quarter of the period of the control signal. Then the period of the control signal can affect the actual output phase of the gyrator. The period of the input signal and control signal is *Tm*, *Tc*, respectively. Without considering the loss of the transmission, the scattering matrix of the gyrator considering the phase change of the buffering unit is:

$$S_{gyrator} = \begin{bmatrix} 0 & \exp(j(\frac{Tc}{4Tm}\pi + 2\varphi_{A \text{ or } B})) \\ \exp(j(\frac{Tc}{4Tm}\pi + \varphi_A + \varphi_B)) & 0 \end{bmatrix} \tag{5}$$

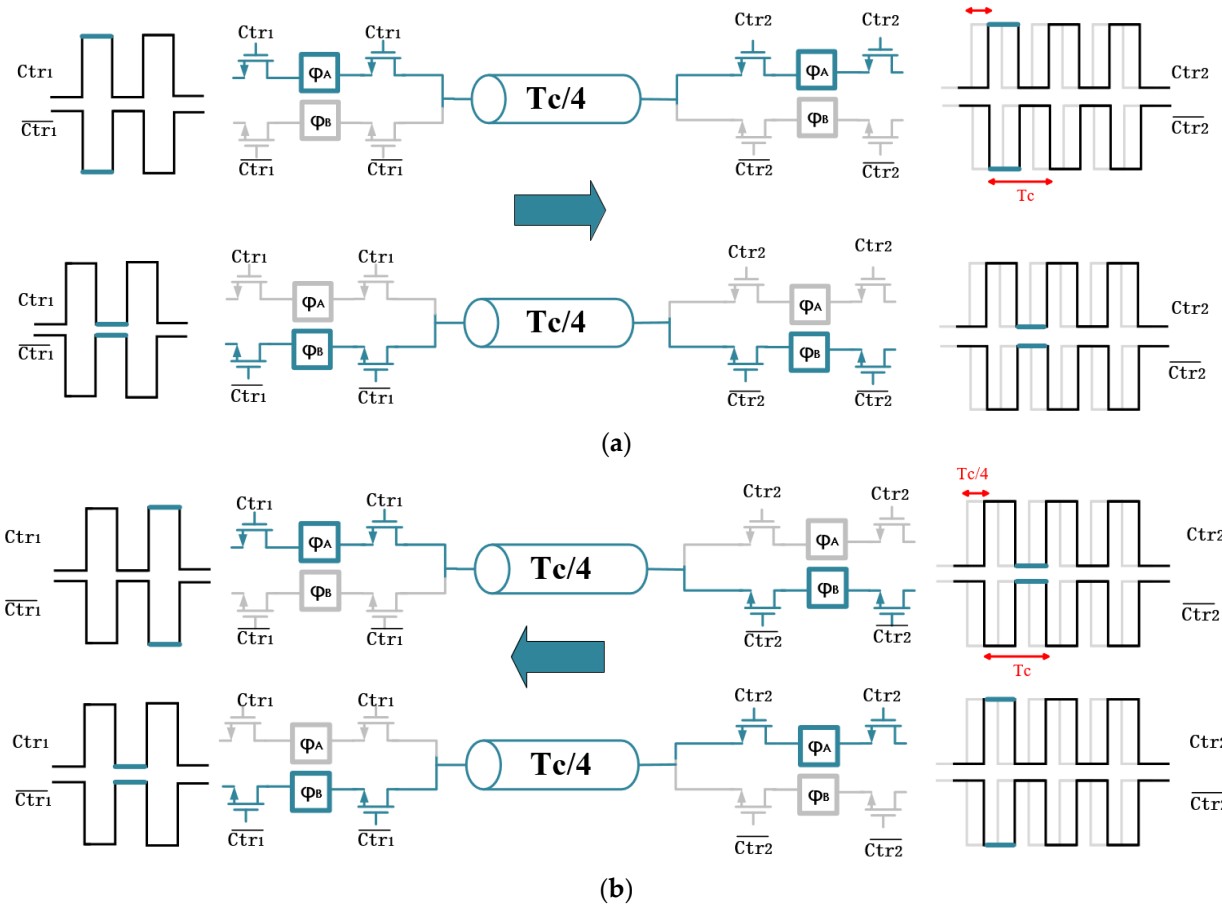

**Figure 4.** (**a**) Propagating transmission. (**b**) Opposite transmission.

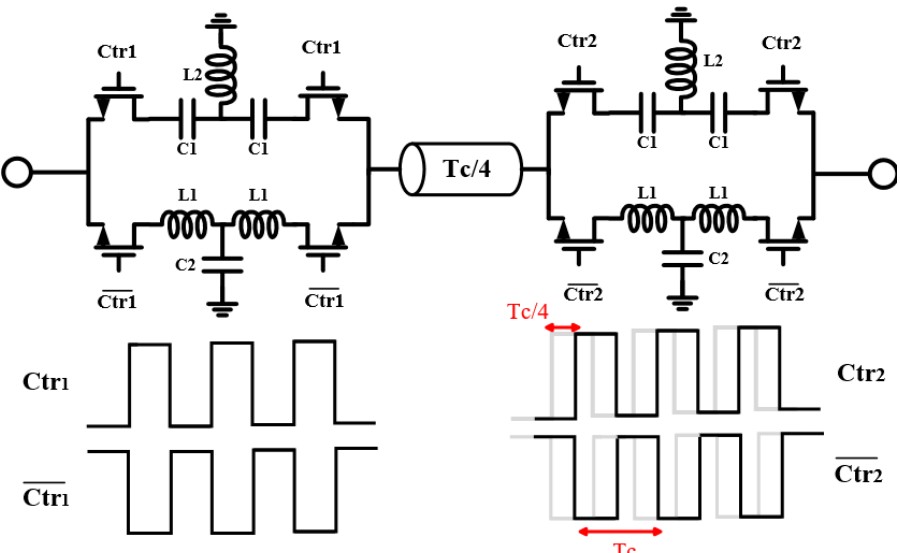

**Figure 5.** The actual circuit structure of the gyrator.

$\varphi_{A \ or \ B}$ in Equation (5) indicates $\varphi_A$ or $\varphi_B$, which is a time-varying quantity. In fact, $\varphi_B$ and $\varphi_A$ differ by 180° at the center frequency, so the value of item $S_{12}$ in Equation (5) is constant. When the frequency of the input signal is changed, the phase difference between $S_{12}$ and $S_{21}$ can still be maintained at 180° within a considerable bandwidth. In addition, the value of $Tm$ also affects the phase change of the gyrator.

Figure 6 shows the phase difference between the forward and the opposite transmissions. The designed TVPM is used to compose the circuit of the gyrator. The differential phase shifter structure can be maintained in a considerable bandwidth. The phase difference caused by forward and reverse transmissions within a certain bandwidth is always maintained at 180°. When the center frequency is 1 GHz, the phase difference of 180° is maintained within the bandwidth of 400 MHz (−3 dB).

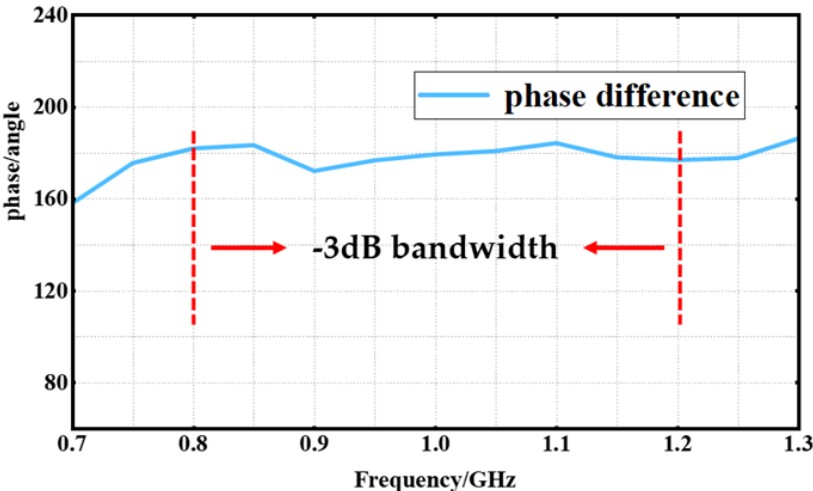

**Figure 6.** The phase difference between the forward and the reverse transmissions.

In fact, when $T_m/T_c$ is odd, the phase change of the output of the two ports is ±90° respectively. Figure 7 shows the phase change caused by the periods of the control signals. When the value of $T_m/T_c$ is 5, the scattering matrix of the gyrator in a time-harmonic form satisfies the following relationships: $S_{12} = j$, $S_{21} = -j$.

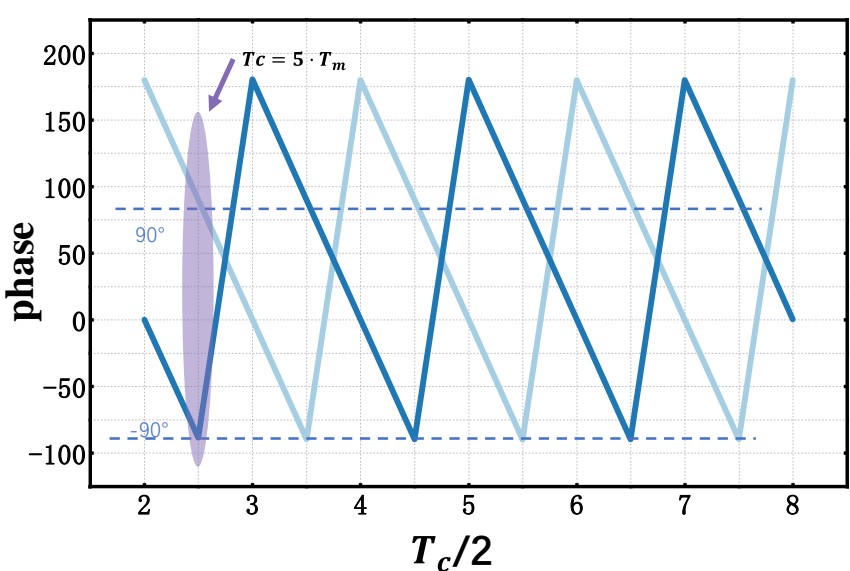

**Figure 7.** The phase change is caused by the periods of the control signals.

### 2.3. Conversion Loss

Figure 8a shows the phase modulation of the single-frequency sine wave in the time domain by the TVPM with the 180° differential phase shifter. It is worth noting that during the process of switching in the TVPM, the phases of the carrier waves can be modulated by the capacitors and inductors, and time is needed to stabilize the carrier output. Such a loss cannot be avoided during the phase conversion process. A partial carrier signal could

be converted into the clutter of other frequencies, as shown in Figure 8b. Therefore, the transmission process of the periodic carrier signal needs to be long enough to reduce the loss of carrier signal [19].

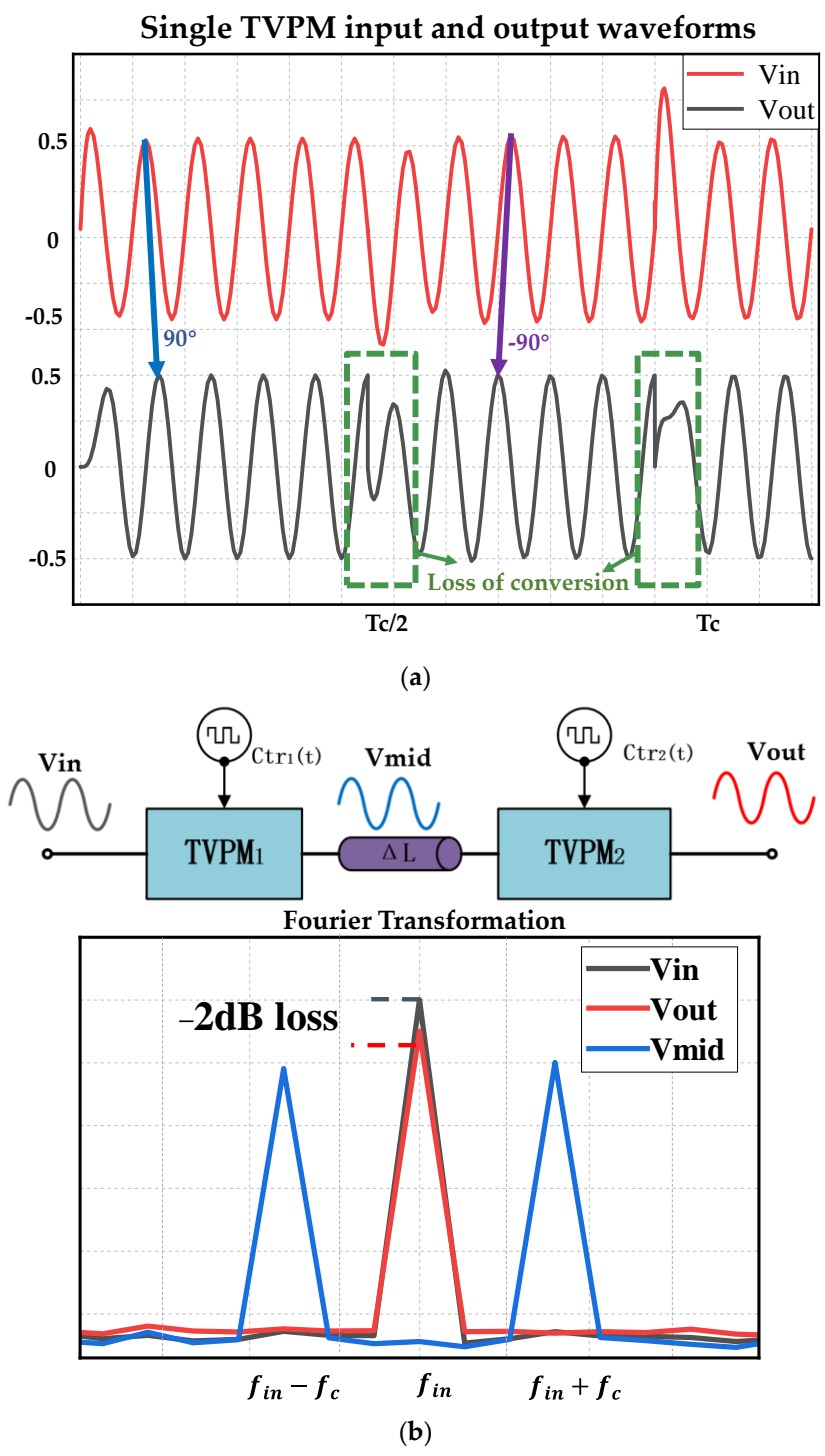

**Figure 8.** (**a**) Switch and phase shifter loss. (**b**) Frequency loss.

In addition, according to Reference [16], the switching channels correspond to the mixer operation. With the frequency of the input signal $f_{in}$, and the control signal $f_c$. Figure 8b shows the frequency losses. When the input signal (black curve) passes through the first TVPM, the frequency is converted, and two frequency signals of $f_{in} - f_c$, $f_{jn} + f_c$ are generated for output, namely the signal of the intermediate stage (blue curve). The

non-reciprocal device requires two different TVPMs. The input signal entering into the second TVPM is converted into other frequencies after passing through the first TVPM and buffer unit. Therefore, the bandwidth of a single TVPM seriously affects the amplitude of the output signal (red curve). The TVPM demonstrated in the paper is a differential 180° phase shifter, the bandwidth of −3 dB is 0.8–1.2 GHz, and the signal with frequency at the boundary of this bandwidth has a large suppression loss (−3 dB) compared with the central frequency of 1 GHz. When the signal passes through two TVPMs, the signal amplitude at the center frequency is weakened (−2 dB). This can be mitigated by extending the period of the control signal or increasing the bandwidth of the phase shifter unit.

In general, the switching process and phase shifting process could cause partial signals to be converted to other frequency components and cannot be resumed. In addition, during the mixing process, the intermediate signal is also suppressed when the frequency reaches the bandwidth boundary of the phase shifter. These two conversion losses cannot be avoided but can be mitigated by increasing the period of the control switch signal and increasing the bandwidth of the phase-shifting unit.

In the ideal situation, the time loss can be neglected, −and the period of the control signal is $T_C$ . The phase control function of TVPM in the time domain can be written as:

$$\varphi(t) = \begin{cases} e^{j\frac{\pi}{2}} & , nTc \leq t < (n+0.5)Tc \\ e^{-j\frac{\pi}{2}} & , (n+0.5)Tc \leq t < (n+1)Tc \end{cases} \tag{6}$$

Table 2 lists the comparison with the other published results [15,16,19]. For scientific and effective exploration, the data listed in [15,16], and this paper are all simulation results. The data show that the proposed differential phase shifter structure can be used to obtain a larger bandwidth than that in Reference [15]. Compared with Reference [16], the overhead area of the circuit is reduced by one degree of freedom.

**Table 2.** Comparison between different methods to achieve the gyrator.

| Reference | [19] [1] | [15] | [16] | This Work |
|:---:|:---:|:---:|:---:|:---:|
| Architecture | Magnetic ferrite | N-path filter | Conductivity mod. | TVPM |
| ANT port | Single end | Single end | Differential end | Single end |
| Frequency | 1 GHz | 0.75 GHz | 25 GHz | 1 GHz |
| BW(−3 dB) | 100 MHz | 40 MHz | Infinite [2] | 400 MHz |
| Insertion loss | −1 dB | −2 dB | −0.5 dB | −1.7 dB |
| C/M ratio [3] | N/A | 1 | 3 | 5 |

[1] Estimate based on the ferrite phase shifter. [2] Limited by the cut-off frequency of the transmission line. [3] Center frequency/modulation frequency.

## 3. Circulator Implementation

A circulator device is implemented by the above-mentioned gyrator and multiple reciprocity devices [19]. As shown in Figure 9, the conduction resistance of the switch is set to 0.1 Ω. The center frequency of the circulator is 1 GHz.

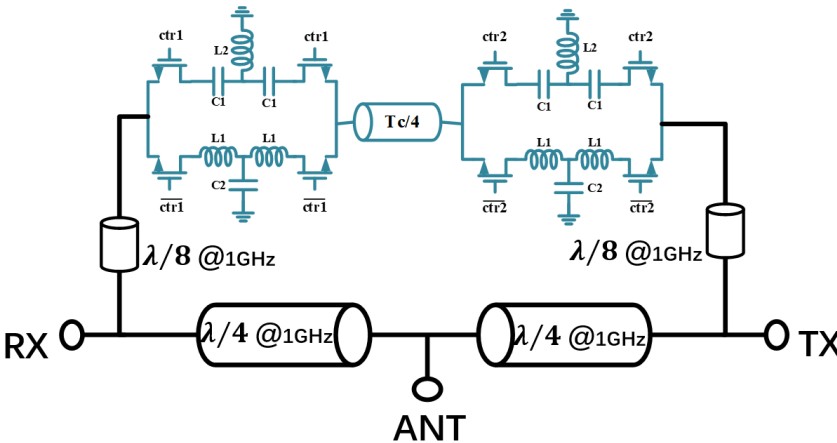

**Figure 9.** Circuit of the designed circulator based on the TVPM.

The transmission line is an ideal one, with the characteristic impedance is 50 Ω. The delay unit is based on the transmission line, with a delay time of 1.25 ns. The RX, ANT, and TX ports are denoted by port1, port2, and port3, respectively. According to the definition of the scattering matrix, to obtain the specific value of the scattering matrix of the three-port system, it is necessary to ensure that only one port is excited and the other two ports are perfectly matched without reflecting waves.

$$S_{ij} = \frac{V_i^-}{v_j^+} \mid _{v_k^+=0, k \neq j} \tag{7}$$

For example, it is necessary to ensure that port2 and port3 are matched to obtain the values of the three scattering matrix elements under the excitation of port1. The time-domain waveform of the voltage values of the three ports is obtained by using the sine wave excitation of port1, and the response amplitude and phase of each port at the center frequency are obtained by Fourier transform. Using Equation (7), $S_{11}$, $S_{21}$, $S_{31}$ can be obtained.

Figure 10 shows the method of calculating the S-parameters. The matching of the port impedance has a great impact on the performance of the circulator. Theoretically, the impedance transparency of the ±90° rotary device [20] can be used to calculate the matching impedance of each port. During the process of frequency conversion, it cannot be guaranteed that the impedance of each port will continue to match all of the time, so there are still some errors.

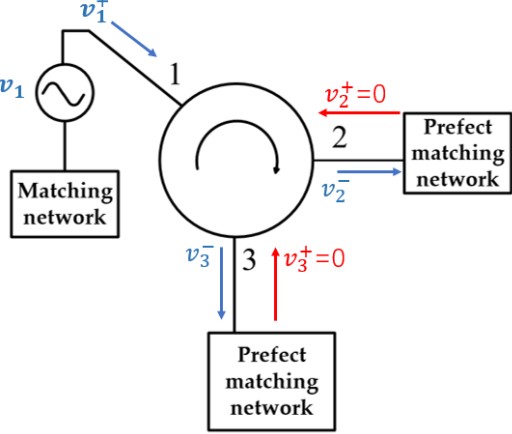

**Figure 10.** Method of calculating S-parameters.

Figure 11 shows the simulation results of the time domain and scattering parameters. S11, S21, and S31 parameters are shown in Figure 11a. S12, S22, and S32 parameters are shown in Figure 11b. Figure 11c shows S13, S23, and S33. Based on the simulation results of all matrix elements of the scattering matrix, it can be demonstrated that the directional transmission of the circulator and the isolation performance between ports can be realized. The insertion loss at the center frequency is −1.7 dB, and the isolation is −20 dB with the antenna as the excitation port. The actual insertion loss of the circulator is smaller than −3 dB. The good insertion loss performance can be attributed to the fact that the gyrator is placed between the incoming transmission lines leading to the reuse of part of the original reflected loss [21].

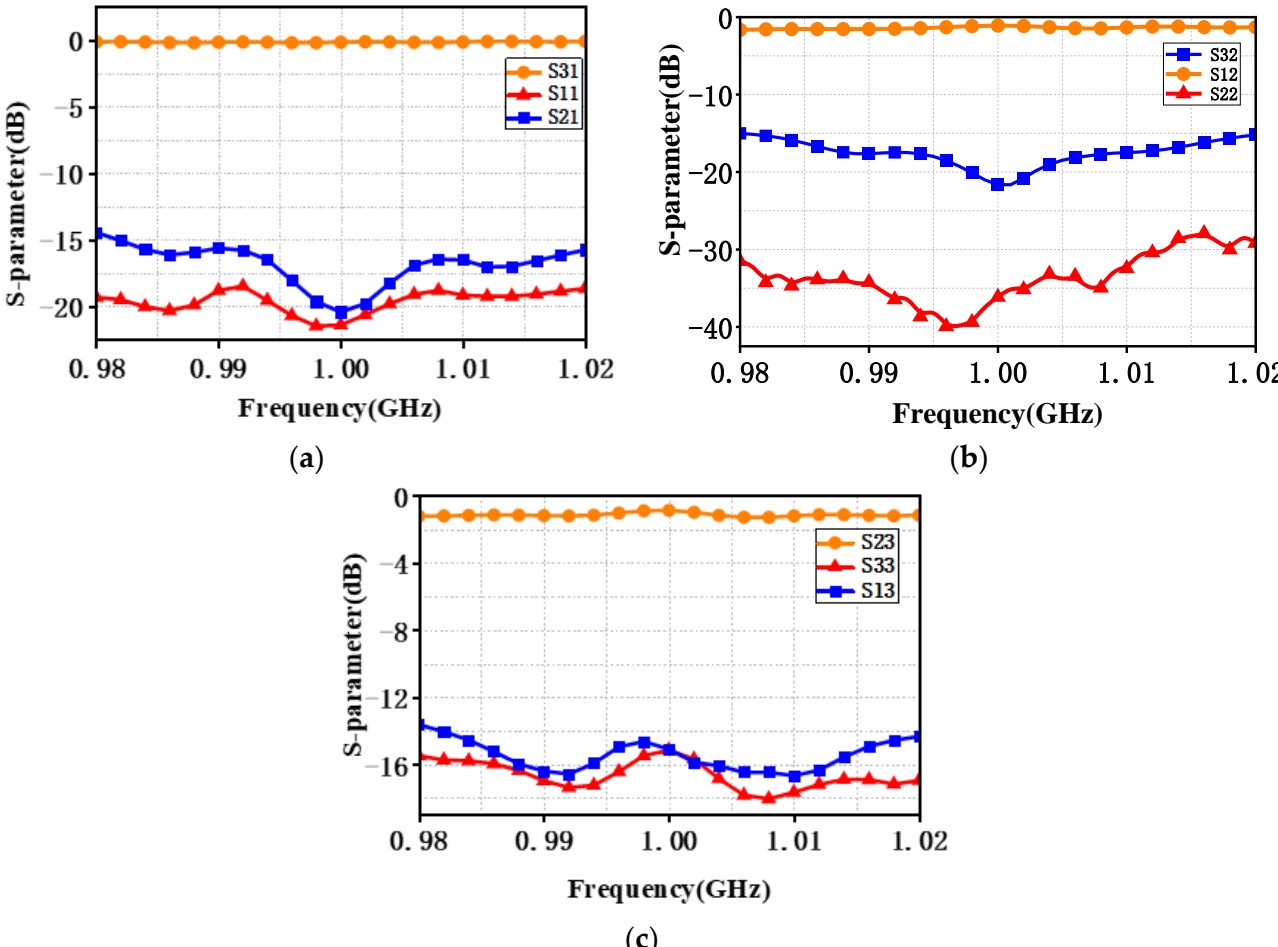

**Figure 11.** S-parameters of the designed circulator. (**a**) S11, S21, and S31. (**b**) S12, S22, and S32. (**c**) S13, S23, and S33.

The scattering matrix structure calculated by the above simulation method still has errors. However, the method of time domain conversion to the frequency domain is still a relatively simple and intuitive approach.

## 4. Discussion

The time-varying modulation method allows for the integration of the circulator into the chip, which is an unparalleled advantage over conventional circulators with external bias fields. Although the devices that constitute the time-varying modulated circulators are passive devices, transistors, etc., which can be realized by modern technology, it is a very critical engineering problem to make the control signal generated by the oscillation

circuit as accurate as possible and to make the insertion loss as small as possible in terms of technology, which requires further research and experimental demonstration.

## 5. Conclusions

In the paper, the concept of a time-varying phase modulator (TVPM) is proposed. By taking advantage of the characteristic that the time-varying system can break the symmetry of time inversion, the general topological structure of the gyrator formed by the time-varying phase modulation method is proposed. The gyrator is composed of common differential phase shifters and different electrical control signals. Simulation results show that the proposed gyrator has a relatively high bandwidth and relatively compromised insertion loss. The circulator is implemented by using the gyrator and the appropriate transmission line. The simulation results show that the average insertion loss of the circulator is $-2$ dB, and the average isolation is $-18$ dB. Most of the insertion loss comes from the differential phase shifter, in which the initial part of the carrier is changed into another frequency during the phase conversion. The aim of the paper is to provide a compromised solution to the N-path filter and the conductivity modulation method of the circulator.

It can be concluded that the circulator in this paper has good performance. The TVPM circuit is a common 180° differential phase shifter structure. In the future, it could have good development prospects in the field of high-speed communication.

**Author Contributions:** Methodology, X.J.; Formal analysis, X.J.; Writing—original draft, X.J.; Writing—review & editing, Y.J.; Supervision, Y.J. All authors have read and agreed to the published version of the manuscript.

**Funding:** This work was supported by the National Natural Science Foundation of China, under grant No. 61774078.

**Institutional Review Board Statement:** The study didn't require ethical approval.

**Informed Consent Statement:** Not applicable.

**Data Availability Statement:** The data that support the findings of this study are available from the corresponding author upon reasonable request.

**Conflicts of Interest:** The authors declare no conflict of interest.

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
