# Peer review of "Non-Magnetic Circulator Based on a Time-Varying Phase Modulator"

_applsci, doi:10.3390/app13010512_

Round 1

Reviewer 1 Report

This paper considers a circulator device for communication systems, based on time-varying phase modulation. This is a well-written and interesting paper, describing a new concept of circulator. In my opinion, it could be accepted for publication in Applied Sciences.

My only comment is related to the signals of Ctr1 and Ctr2 appearing in figures 4 and 5, which meaning is not so obvious. I wonder if they could be improved and/or simplified.

Reviewer 2 Report

Dear Editor,

In this paper, the authors introduced a Non-Magnetic Circulator Based on Time-Varying Phase Modulator. However, its innovation is not outstanding enough. In addition, there are only theoretical derivations and simulations in the manuscript. Due to the lack of experimental verification, the simulation results are unconvincing. Please see below for detailed comments.

1.       In this paper, simulation test is carried out, but experiment is not carried out. I mean, the experiment should be carried out.

2.       What’s the main novelty of this manuscript?

3.       The authors should compare the proposed scheme with other technology in introduction.

4.       I think the whole paper needs to remove typos and grammatical errors.
